# Data-driven insights into neighborhood adherence to cancer prevention guidelines in Philadelphia

**Tesla D. DuBois**[1,2], **Kari Moore**[3], **Heather Rollins**[3], **John Silbaugh**[1], **Kristen A. Sorice**[1], **Shannon M. Lynch**[1,4]*

1 Division of Cancer Prevention and Control, Fox Chase Cancer Center, Temple University Health System, Philadelphia, Pennsylvania, United States of America, 2 Geography Department, Temple University, Philadelphia, Pennsylvania, United States of America, 3 Urban Health Collaborative, Drexel University, Philadelphia, Pennsylvania, United States of America, 4 Center for Biostatistics and Epidemiology, Lewis Katz School of Medicine, Temple University, Philadelphia, Pennsylvania, United States of America

* shannon.lynch@fccc.edu

## Abstract

**Data Availability Statement:** This study relies exclusively on third-party data contributed from the Philadelphia Health Management Corporation, the U.S. Census Bureau American Community Survey,

### Background

Forty percent of new cancer cases in the United States are attributed to modifiable risks, which can be influenced by the built environment. Recent cancer prevention guidelines include recommendations for making communities conducive to healthy living. Focused on the city of Philadelphia, the present study aims to 1) evaluate neighborhood-level adherence to cancer prevention guidelines by developing two novel indices and 2) identify factors driving low compliance in neighborhoods with high cancer mortality.

### Methods

Philadelphia neighborhoods were compared to the city overall on ten cancer prevention recommendations. Comparison scores informed two indices: one focused on the American Cancer Society's guidelines for Physical Activity, Nutrition, and Smoking, and the other focused on Healthy People 2030's guidelines for Prevention Services. Indices were mapped by neighborhood and compared to cancer mortality. Where low adherence overlapped with high cancer mortality, the recommendations driving low compliance were identified.

### Results

Distinct geospatial patterns were observed in adherence to guidelines, and while drivers of low adherence varied by neighborhood, general trends emerged in different areas of the city. Concerning Physical Activity, Nutrition, and Smoking Guideline adherence, some areas appeared to be more influenced by the built environment, while others were impacted by specific behavioral risk factors such as excessive alcohol consumption. Preventive Service recommendation adherence was driven in some parts by self-reported poor health and, in others, low cancer screening rates and a high physician-to-resident ratio. In neighborhoods

the Philadelphia Police Department, the City of Philadelphia, the United States Department of Agriculture, Walk Score®, and the Pennsylvania State Cancer Registry. Descriptions of variables used in this analysis, the year, and data product from which they are pulled are detailed in the supporting materials (S1 Table) of this manuscript. Additionally, most aggregated neighborhood-level estimates used in this study as well as the neighborhood boundary files and code associated with this manuscript have been made available by the authors on a public GitHub repository, which can be found here: https://github.com/LynchLabFCCC/DuBois-et-al.-PLOS-ONE-code. Only Walk Score® data cannot be redistributed based on our data use agreement, however, it can be requested by interested parties here: https://www.walkscore.com/professional/research.php. Base maps were not utilized in the production of the maps produced for this manuscript.

**Funding:** This work was supported by funding by the American Cancer Society MRSG CPHPS -130319 to SML. The contributions of KM and HR were supported by the Lazarex Cancer Foundation through funds provided by the Silicon Valley Community Foundation. We thank them for their support but acknowledge that the findings and conclusions presented are those of the author(s) alone, and do not reflect the opinions of the funders. Neither funder played a role in study design, data collection and analysis, decision to publish, or preparation of the manuscript.

**Competing interests:** The authors have declared that no competing interests exist.

where poor guideline adherence overlapped with high cancer mortality, the built environment emerged as a potentially important factor.

## Discussion

This study considers the importance of the built environment in influencing adherence to cancer prevention guidelines. Policymakers and public health officials can use this information to prioritize interventions for neighborhoods with low guideline adherence and high cancer burden and tailor interventions to focus on indicators of low guideline adherence.

## Introduction

Although cancer mortality has steadily decreased in the United States for the past two decades, cancer remains the second leading cause of death nationally, with nearly 612,000 cancer deaths estimated in 2024 [1]. A 32% drop in cancer deaths between 1991 and 2019 has been attributed to reduced risk behaviors (primarily smoking) and advances in screening and treatment [2]. However, approximately 45% of cancer deaths are attributable to modifiable risks, including smoking, excess body weight, alcohol intake, poor diet, and lack of physical activity, which if addressed, could improve cancer prevention [3]. An individual's surroundings, including both the social and built environment of the neighborhood in which they live, could influence behaviors associated with cancer. For example, residents may be less likely to exercise and subsequently improve body weight if they live in an area with limited access to safe recreation space [4]. Therefore, neighborhood-level factors are increasingly becoming an important consideration in cancer prevention.

Recognizing the potential to decrease the cancer burden by addressing modifiable risks and behaviors, the American Cancer Society (ACS) set a goal to reduce overall cancer mortality by 40% between 2015 and 2035, primarily by targeting behavioral factors. Meeting this goal would result in 1.3 million fewer cancer deaths in the United States between 2020 and 2035 [5]. The ACS has published an updated version of its Physical Activity and Nutrition Guidelines to support this goal [6]. These guidelines are developed by a panel of national experts and reflect current scientific evidence linking specific behaviors (such as diet and physical activity) to cancer. Recognizing the effect of the built environment on risk behaviors, the most recent version of the ACS guidelines, published in 2020, also includes neighborhood-level recommendations, such as improving access to health care, healthy foods, and safe spaces to exercise [7].

In conjunction with the ACS guidelines, Healthy People 2030 (HP 2030), published by the United States Department of Health and Human Services, Office of Disease Prevention and Health Promotion, is a set of evidence-based objectives aimed at improving health and well-being of US residents over the coming decade. The goals of HP 2030 include targets for cancer screenings and access to care, in line with the ACS guidelines [8]. Together, the ACS Physical Activity and Nutrition Guidelines and the HP 2030 provide objective and measurable standards for decreasing cancer risk through modifiable factors, evaluated in the context of neighborhood environment. However, the majority of studies to date have evaluated adherence to cancer prevention and screening guidelines only at the individual level and have not measured adherence in the context of the built environment or neighborhood-level factors [9]. Considering how guideline adherence varies by neighborhood could be useful, as mapping neighborhood adherence could inform where and what type of tailored interventions may be utilized to improve cancer prevention. This approach—which utilizes area-level, geospatial analysis to

assist with allocating often limited resources for intervention efforts to the highest burdened communities—is referred to as Precision Public Health [10].

The present study is focused on the city of Philadelphia, where the cancer mortality rate of 190 per 100,000 residents is much higher than the State and National rates (164 and 144 per 100,000, respectively) [11, 12]. Evaluating adherence to cancer prevention guidelines at a neighborhood level in Philadelphia can help inform those working in cancer prevention to address the city's high cancer burden. Thus, our first goal was to assess which Philadelphia neighborhoods are adhering to ACS and HP 2030 guidelines by evaluating the prevalence rates of behavioral and built environmental variables used to define each guideline recommendation. Second, we developed novel neighborhood-level adherence composite scores for each recommendation, including the development of a new adherence score for evaluating the built environment. These composite scores were then compared to neighborhood cancer mortality rates to determine communities most in need, supporting a Precision Public Health approach for cancer prevention efforts in Philadelphia.

## Methods

The process to create and evaluate adherence to ACS and HP 2030 recommendations by Philadelphia neighborhood was multistep and involved multiple data sources, including those utilized by our prior work on Neighborhood Health Rankings in Philadelphia [13]. This research was covered under Fox Chase Cancer Center IRB # 18–9020. Informed consent was waived as the study relied exclusively on publicly available existing area-level data.

### Neighborhood definitions

We defined neighborhood boundaries within Philadelphia according to aggregated census tracts, as specified in the Neighborhood Health Rankings in Philadelphia Report (S1 Fig). Neighborhoods consist of groups of 4 to 16 census tracts with a combined median population of approximately 33 thousand residents. An improvement over using a decades-old administrative boundary such as zip codes, local knowledge of Philadelphia was considered in the creation of these neighborhoods, resulting in a geographic unit that more closely matches residents' perception of neighborhoods in the present day. There were a total of 46 neighborhoods created in this analysis (S1 Fig) [13] in 8 regions of the city (S2 Fig). Supplemental maps of labeled neighborhoods (S1 and S2 Figs) were produced in ArcGIS Pro 3.0.0.

Next, we generated summary composite scores for each recommendation by neighborhood, before rolling them into two overall summary adherence scores, one for ACS Physical Activity, Nutrition, and Smoking Guidelines and another for Healthy People Prevention Services Guidelines (Fig 1). Finally, in our spatial analysis, we mapped overall adherence scores by Philadelphia neighborhood (Fig 2) and compared them to cancer mortality (Fig 3). We then identified which recommendations are driving low adherence in the neighborhoods with the highest cancer mortality rates (Tables 3 and 4). Maps of individual measures and mortality (S1 Fig), adherence scores (Fig 2) and overlap between low adherence and high mortality (Fig 3) were produced in the R Language and Environment for Statistical Computing (version 4.4.0) [14], using the "ggplot2" [15] and "ggpubr" [16] packages.

### Guidelines and recommendation measures

**Guidelines.** The ACS Physical Activity and Nutrition Guidelines, accompanied by behavioral recommendations regarding cigarette smoking are detailed in Table 1. Table 2 presents the HP 2030 guidelines for preventive services, as well as the ACS screening guidelines. These guidelines are summarized as numbered recommendations. Our study specifically focused on

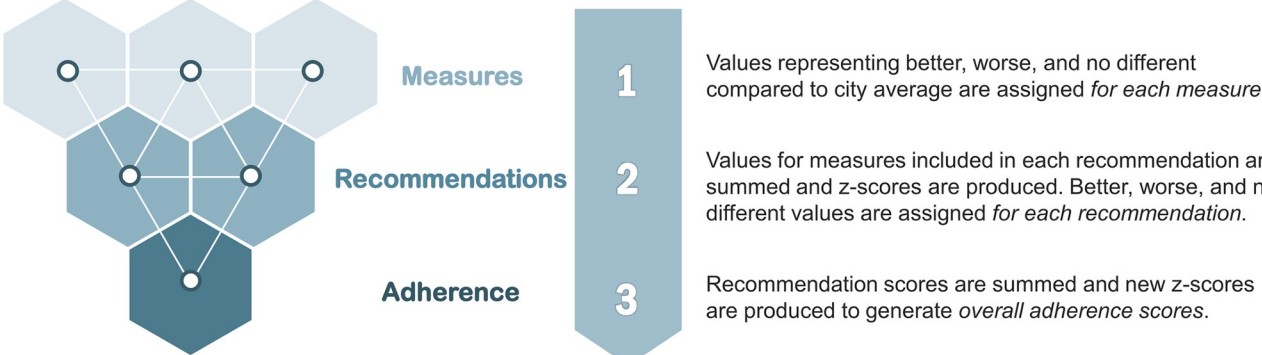

**Fig 1. 3-step process of rolling individual measures into an overall guideline adherence score.** (1) Values representing better, worse, and no different compared to city average are assigned *for each measure*. (2) Values for measures included in each recommendation are summed and z-scores are produced. Better, worse, and no different values are assigned *for each recommendation*. (3) Recommendation z-scores are summed and new z-scores are produced to generate *overall adherence scores*.

guidelines for which we had accessible measures to represent each recommendation. Therefore, recommendations from the ACS regarding prostate or lung cancer screening, for example, were not included.

Tables 1 and 2 provide a comprehensive overview of each recommendation, including the name of the recommendation (column 1), the measure or measures used to assess them

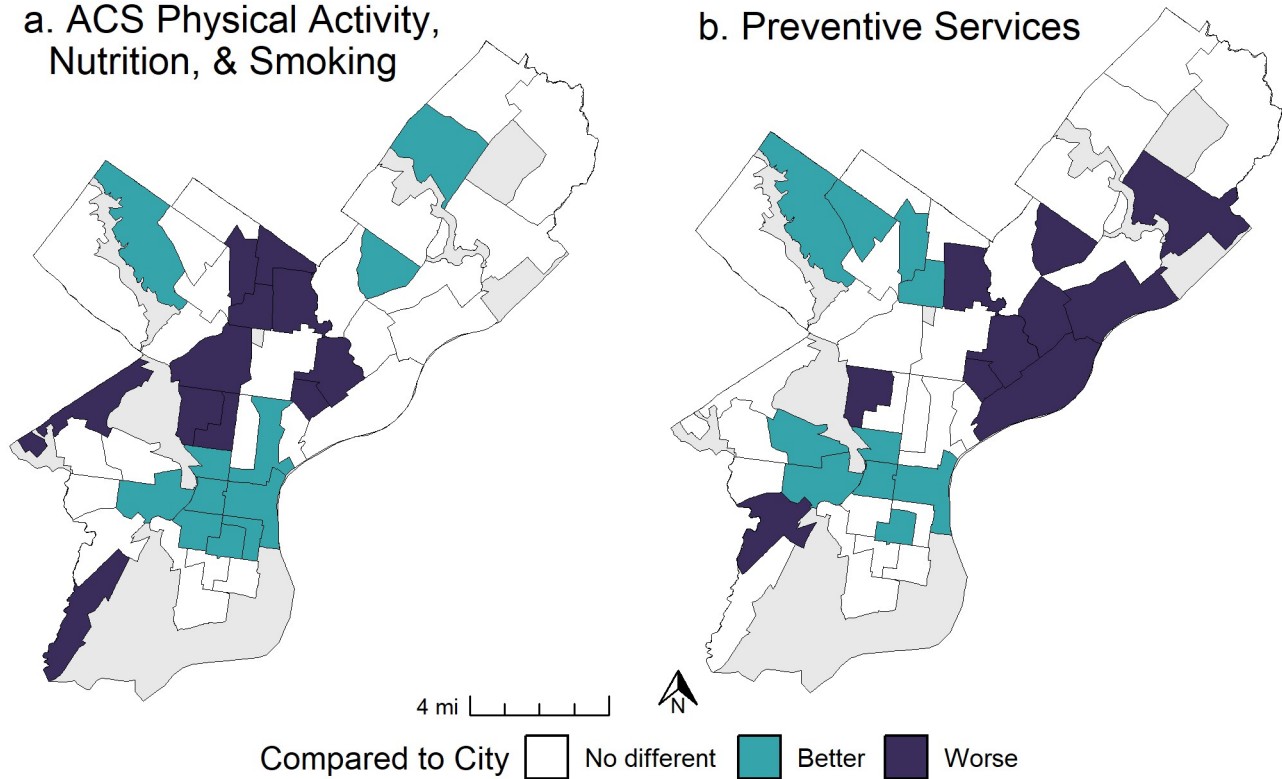

**Fig 2. Overall adherence scores by Philadelphia neighborhood.** (a) Adherence to ACS Physical Activity, Nutrition, & Smoking Guidelines, showing neighborhoods as either worse, better, or no different than overall adherence in the city. (b). Adherence to Preventive Services Guidelines, showing neighborhoods as either worse, better, or no different than overall adherence in the city.

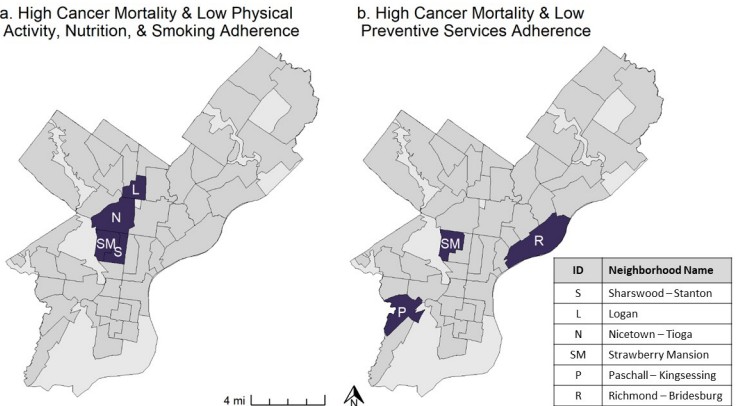

**Fig 3. Overlap between worst guideline adherence and highest cancer mortality.** (a) High cancer mortality neighborhoods that also have low adherence to ACS Physical Activity, Nutrition, & Smoking Guidelines. (b). High cancer mortality neighborhoods that also have low adherence to Preventive Services Guidelines.

**Table 1. ACS physical activity, nutrition and smoking guidelines.**

| Recommendation | Measure(s) | US Target or US Average | Mean (SD) of Neighborhoods (n = 46) | "Better" Quartile Range | "Worse" Quartile Ranges |
|---|---|---|---|---|---|
| #1- Achieve and Maintain a Healthy Weight Throughout Life | The percent of residents with a BMI greater than or equal to 30 kg/m$^2$ | HP Target: <36.0% [17] | 32.6(5.12) | 19.3–30.0 | 36.3–39.6 |
| #2- Be Physically Active | The percent of residents exercising 30 minutes at least 3 times/week AND meeting the non-sedentary guidelines of sitting while watching a screen during leisure time for less than 5 hours per day | HP Target: 52.9% of adults getting 150 minutes of exercise per week [18] | 36.4(4.7) | 38.5–51.7 | 29.4–33.1 |
| #3- Eat a Healthy Diet, with an Emphasis on Plant Foods | Percent of residents consuming 5+servings of fruit/veggies daily | US: 10–12% [19] | 10(1.9) | 11.1–15.1 | 7.3–8.6 |
| | The percent of residents who drank 1 or more sugar-sweetened beverages per day | US: 50% [20] | 30.21(5) | 18.3–26.6 | 34.9–38.5 |
| #4- Limit Intake of Alcohol | Percent of residents who are binge drinking (4+ drinks for women or 5 + drinks for men in two hours) | HP Target: <25.4% [21] | 35.63(4.3) | 28.2–32.3 | 39.0–44.3 |
| #5- Stay Away from Tobacco | Percent of residents that are current smokers | HP Target: <6.1% [22] | 19.51(4) | 10.5–16.9 | 22.5–26.4 |
| #6- Increase access to affordable, nutritious foods | The percent of residents who are further than 1/2 mile from the closest supermarket | US: 40% live more than a mile from grocery [23] | 21.4(17.7) | 0–7.7 | 35.9–65.4 |
| #7- Provide safe, enjoyable, and accessible environments for physical activity in schools and workplaces, and for transportation and recreation in communities | Percent of households which own at least one vehicle | US: 91.7% [24] | 68.9(13.1) | 80.8–94.2 | 46.7–57.6 |
| | The average walk score (out of 100) based on intersection density, residential density, and accessibility of amenities such as grocery stores, parks, and restaurants | US: 48 (walk score in cities with at least 200k residents) [25] | 75.2(17.3) | 88.9–98.6 | 33.3–65.2 |
| | The rate of violent crimes (murder/ homicide, aggravated assault, robbery, and rape) per 100,000 people | US: 379 [26] | 944(56.8) | 141–571 | 1,293–2,449 |
| | The percent of residents who responded "yes" to having a park in their neighborhood that they felt comfortable visiting | Not Available | 75.8(8.7) | 81.9–93.1 | 56.5–69.7 |

**Table 2. Preventive service guidelines.**

| Recommendation | Measure(s) | US Target or US Average | Mean (SD) of Neighborhoods | "Better" Quartile Range | "Worse" Quartile Ranges |
|---|---|---|---|---|---|
| HP: Increase the proportion of adults who self-report good or better health | The percent of residents who self-reported a health status of good or higher | 2020 Target: 78.8% [27] | 78.3(5.76) | 82.8–89.5 | 63.3–74.7 |
| HP: providing population-based primary prevention services | Population per primary care provider | 1,320 [28] | 1,543.1(851.3) | 312.2–966.8 | 2,144.7–3,611.3 |
| ACS: Screening | The percent of residents recommended based on age who received a colonoscopy within the past 10 years | 68.3% [29] | 72.8(4.7) | 76.0–81.0 | 60.5–70.2 |
| | The percent of residents recommended based on age/sex who received a mammogram within the past 2 years | 80.3% [30] | 84.0(4) | 86.3–91.8 | 72.9–81.8 |
| | The percent of residents recommended based on age/sex who received a pap smear test within the past 3 years | 79.2% [31] | 85.5(2.6) | 87.6–90.5 | 80.1–83.4 |

(column 2), and the United States Target (as defined by Healthy People 2030) or, in cases where the target is undefined, the national average for the measure (column 3). The tables also display the mean and standard deviation of neighborhood values (column 4), the range of values considered "better" than the city overall based on neighborhood inclusion in the most favorable quartile (column 5), and the range of values considered "worse," representing the most unfavorable quartile (column 6).

**Data sources for recommendation measures.** Data collection efforts aimed to identify measures that 1) could serve as a measure of guideline recommendations of interest (see Tables 1 and 2), and 2) were available at a scale that could be aggregated to the neighborhood level (see Neighborhood Definitions). Multiple data sources were used to evaluate guideline recommendations (S1 Table). The Southeastern Pennsylvania Household Health Survey (SEPAHHS), a phone survey of adults in the greater Philadelphia region conducted from 2000–2018 by the Public Health Management Corporation (PHMC), contributed several measures related to recommendations #1–5 & #7 of the ACS Physical Activity and Nutrition Guidelines and recommendations #1 & #3 of the Preventive Services Guidelines (S1 Table). Data for the measure for recommendation #6 was provided by USDA Food Access Research Atlas and had been previously aggregated to the neighborhood-level by the authors for the Close to Home Report, 2019. Similarly, recommendation #7 included neighborhood estimates from the Close to Home Report based on data provided by the Philadelphia Police Department and by Walk Scor®. Recommendation #7 also included data from the American Community Survey (2018) provided by the Census Bureau. Finally, cancer mortality data from the Pennsylvania Department of Health Cancer Registry were based on 2012–2016 Pennsylvania death certificates that identify cancer as the underlying cause of death. S1 Table contains detailed information on each included measure (including measure definition, data source, year, and method of aggregation to the neighborhood level).

**Generating measures for guideline recommendations.** Methods for aggregating measures from individual level survey measures to produce neighborhood level prevalence rates for each guideline recommendation varied depending on the structure of the initial dataset (S1 Table). After appropriate aggregation, an overall prevalence rate for the city was calculated for each measure and recommendation as the mean of all neighborhood rates for the given measure.

## Adherence score generation

Adherence scores were generated for a) each recommendation that had more than one measure representing the recommendation b) for ACS Physical Activity, Nutrition, and Smoking Guidelines overall c) for Healthy People Prevention Services Guidelines overall. For each recommendation, individual measures related to specific recommendations were rolled into a neighborhood guideline adherence measure using a three-step process (Fig 1).

**Step 1: Assign ordinal scores of better/worse/no different for *each measure*.** Each neighborhood is assigned a categorical score of "above," "below," or "no different" from the city overall for each of the 18 included measures. Difference scores were assigned as above or below the city if the value fell into the lowest or highest quartile of all neighborhoods on that measure. The range of values considered above and below for each measure are provided in Table 1.

Reverse coding was conducted, as necessary, based on the directionality of the measure. For example, having a higher-than-average score on the variable related to eating a healthy diet can be interpreted as "better" than average because eating a healthy diet is recommended by the ACS guidelines. However, being above the city average for tobacco use would be considered "worse," as Healthy People 2030 aims to reduce tobacco use. For each of the 18 measures, a value of 1 was assigned to "better," -1 to "worse," and 0 was assigned where neighborhoods were not significantly different from the city average (S1 Table).

**Step 2: Assign z-scores for each recommendation.** Each recommendation is measured using one to four variables (Tables 1 and 2), each with an associated value of -1, 0, or 1, produced in Step 1. For recommendations consisting of more than one measure, we summed the associated variable values within each recommendation. We then calculated z-scores for each recommendation.

**Step 3: Calculate scores for overall adherence.** Recommendation z-scores are summed, and better, worse, and no different categories are applied to the Total Adherence scores based on quartile breaks of z-score sums. Neighborhoods in the first quartile (lowest adherence scores) are considered worse than the average neighborhood. In contrast, those in the fourth quartile (highest scores) have better than average overall adherence to cancer prevention guidelines.

## Statistical/Spatial analysis

Summary statistics, including the mean and standard deviation of the neighborhood prevalence rate of each guideline recommendation is presented in Tables 1 and 2 and compared to average or targeted U.S. prevalence rates for each recommendation according to the HP 2030 report. Adherence to each of the measures included in the seven recommendations related to ACS Physical Activity, Nutrition, and Smoking Guidelines and the three measured Healthy People Prevention Services Guidelines recommendations were presented in a series of maps by the 46 Philadelphia neighborhoods (S3 Fig). Visual inspection of the maps revealed where the city neighborhoods were doing better or worse concerning each measure and cancer mortality compared to the city overall (S3 Fig). Additionally, maps of the overall ACS Physical Activity, Nutrition, and Smoking Guidelines adherence and Healthy People Prevention Services Guidelines adherence indices were presented (Fig 2). Adherence maps were compared to maps of cancer mortality in the city to identify neighborhoods where high cancer mortality overlap with worse adherence to either set of guidelines (Fig 3). To establish whether the

resulting indices are statistically related to cancer burden, a Spearman test of correlation was first conducted before testing for association using a series of univariate linear regressions (S2 Table). Finally, focusing on neighborhoods where high mortality and poor adherence co-occur, recommendations responsible for the poor adherence scores were identified to suggest possible points of intervention specific to a given neighborhood.

## Results

### Summary of guideline recommendations in Philadelphia neighborhoods

Tables 1 (ACS Physical Activity, Nutrition, and Smoking Guidelines) and 2 (Healthy People Prevention Services Guidelines) summarize adherence to each recommendation in Philadelphia overall and record the quantile breaks for each measure. Additionally, they include a column indicating the Healthy People 2030 (HP) target (where applicable) or the US average for the given measure.

Recommendations #1–5 are focused on behavioral risk. The average Philadelphia neighborhood met the HP target for obesity as measured by a Body Mass Index under 30kg/m2 (32.6% vs <36.0%, recommendation 1). Adherence to sedentary guidelines nationally is not available, and the available measures for physical activity at the city and national levels are not equivalent. The HP Target is that 52.9% of adults are getting 150 minutes of exercise per week, however, the data we have on physical activity at the neighborhood level is the percentage of residents that exercise for 30 minutes 3 times a week (for a total of at least 90 minutes) and also meet the non-sedentary recommendation of less than 5 hours daily of sitting with a screen (TV or computer) during leisure time. An average of 36% of Philadelphia residents meet the combined exercise and non-sedentary recommendation, making it unlikely that Philadelphia meets the HP target for physical activity. Consumption of fruits and vegetables are measured separately by the CDC, as opposed to the combined measure used in this analysis. However, it appears that Philadelphia neighborhoods are very similar to national averages (10% combined vs 10% and 12% for vegetable and fruit consumption, respectively, recommendation #3). Philadelphia is lower than the US average for the percentage of adults who drink a sugar-sweetened beverage daily (30.2% vs 50%, respectively, recommendation #3). Comparing Philadelphia to the HP targets, we see that Philadelphia residents consume alcohol and smoke cigarettes at high rates (35.6% vs 25.4% and19.5% vs 6.1%, respectively, recommendations #4 & #5).

Recommendations #6–7 are related to the built environment. A national measure of residents within a half mile of a supermarket is unavailable, however, 40% of US residents live within a mile of a supermarket. A neighborhood average of 21% of residents having a supermarket within a half mile appears to be relatively similar (Recommendation #6). Recommendation #7 includes four measures. Philadelphia has a high walkability score compared to other cities with at least 200 thousand residents (75.2 vs 48 on a scale from 1–100) and a much lower rate of car ownership (68.9% vs 91.7%). While national data on safe parks were not identified, 75% of residents responded yes to having a park in their neighborhood that they feel comfortable visiting. The environment variable with the most measurable contrast to national rates reflects violent crime, with Philadelphia exhibiting a very high rate compared to the national average (944 vs 379 events per 100,000 residents).

Regarding the Healthy People Prevention Services Guidelines, Philadelphia meets the HP 2020 target for self-reported good health (78.3 vs 78.8). Philadelphia has more residents per primary care provider than the national average, likely explained by the urban setting (1,543 vs. 1,320). Philadelphia meets Healthy People 2030 targets for colorectal, breast, and cervical cancer screenings (72.8% vs 68.3%, 84% vs 80.3%, and 85.5% vs 79.2%, respectively).

## Overall adherence by Philadelphia neighborhood

After rolling individual measures (S1 Table) into recommendation-level adherence scores, we calculated overall adherence to the ACS Physical Activity, Nutrition, and Smoking Guidelines and Healthy People Prevention Services Guidelines and mapped the final result (Fig 2a and 2b, S2 Fig). Adherence to guidelines presents distinct geospatial patterns. Multiple neighborhoods in North and Lower Northeast and one each in West and Southwest Philadelphia demonstrate low adherence to ACS Physical Activity, Nutrition, and Smoking Guidelines overall (Fig 2a). Two neighborhoods in North Central, one in Southwest and seven in Northeast Philadelphia had low adherence to preventive service recommendations (Fig 2b). The remaining areas had better or the same adherence to guidelines compared to the city as a whole.

For overall adherence to ACS Physical Activity, Nutrition, and Smoking Guidelines, we see that 7 neighborhoods in the central core of North Philadelphia (Oak Lane Fernrock, Ogontz, Logan, Olney Feltonville, Nicetown–Tioga, Strawberry Mansion, and Sharswood-Stanton), as well as two in the Lower Northeast (Upper Kensington, Juniata Park Harrowgate), one neighborhood in the West (Overbrook Park—Wynnefield Heights) and one in the Southwest portion of the city (Eastwick—Elmwood) perform worse than the city overall. The recommendations and individual measures driving low adherence scores vary by neighborhood, however general trends emerge in different areas of the city (S3 Fig–Maps of individual measures). In the North Central, Lower Northeast, West, and Southwest portions of the city, low guideline adherence is influenced by the built environment recommendation # 7 (likely driven by violent crime), as well as behavioral risk factors including poor diet and lack of physical activity, given that neighborhoods in those areas were worse than the city for these recommendations. The North Central and Lower Northeast portions of the city were also worse than the city on smoking tobacco and while the center and southern portions of the city tend to be better than the city overall on most recommendations, those areas have high rates of excessive alcohol consumption.

For overall adherence to Healthy People Prevention Services Guidelines, areas of low adherence to preventive services emerged predominantly in Northeast neighborhoods (Richmond–Bridesburg, Wissinoming–Tacony, Oxford Circle, Torresdale S.–Pennypack Park, Frankford, Juniata Park Harrowgate, Upper Kensington), as well as two in North (Olney Fentonville and Strawberry Mansion) and one in Southwest Philadelphia (Pachall Kingsessing). In all three regions, high resident to physician ratio coincide with low adherence to Prevention Service guidelines. North Central and Southwest neighborhoods also have low rates of residents reporting good health, and the Northeast portion of the city generally has low cancer screening rates (S3 Fig–Maps of individual measures).

## Adherence score overlap with cancer mortality

We identified neighborhoods with higher cancer mortality rates than the city average (S3 Fig) and where high mortality neighborhoods overlapped with poor guideline adherence scores (Fig 3). Poor adherence to ACS Physical Activity, Nutrition, and Smoking Guidelines is identified in four neighborhoods with increased mortality (Fig 3a), all of which (Logan, Nicetown-Tioga, Strawberry Mansion, and Sharswood-Stanton) are located in North Central Philadelphia. Poor adherence to Healthy People Prevention Services Guidelines overlaps with high mortality in three non-contiguous neighborhoods (Fig 3b), Strawberry Mansion (North Central) and Richmond Bridesburg (Lower Northeast), and Paschall-Kingsessing (Southwest). Both indices were correlated with cancer mortality and a linear regression found that 44.5% of between-neighborhood variation in cancer mortality can be explained by the ACS Physical Activity, Nutrition, and Smoking index alone and 21.8% can be explained by the HP

Prevention Services Guidelines. While the purpose of these indices is not for predicting cancer burden, as they focus only on modifiable behavioral and built environment factors at the area level, this association is important to establish as it speaks to the utility of the indices. As prior research suggests that 45% of cancer deaths are attributable to modifiable factors [3], it is noteworthy that the ACS Physical Activity, Nutrition, and Smoking index can statistically explain the same amount of variation of cancer mortality between neighborhoods using area-level data.

## Identifying recommendations related to poor guideline adherence in high cancer mortality neighborhoods

In Nicetown-Tioga and Strawberry Mansion, poor adherence to ACS Physical Activity, Nutrition, and Smoking Guidelines is driven by recommendations related to maintaining a healthy body weight, being physically active, tobacco use, and providing a safe, enjoyable, and accessible environment (Table 3). Low adherence in Sharswood-Stanton is driven by worse adherence to the recommendations reflective of the built environment (recommendation #7) and alcohol consumption (recommendation #4). Finally, low adherence in the Logan neighborhood is driven by worse body weight, diet, and access to food compared to the city overall. Three of the four neighborhoods with low adherence to ACS Physical Activity, Nutrition, and Smoking guidelines and worse cancer mortality than the city overall are worse than the city overall on recommendation #7, which includes measures related to safety and accessibility, indicating that the built environment may be influential with regard to cancer risk and outcomes.

Car ownership and walkability are both included in recommendation #7 as measures of access. Comparing those measures with each other and with cancer mortality, we see that neighborhoods with low car ownership and low mortality (University City and Center City West) are better than the city average on walkability, which is not true for neighborhoods with low car ownership and high mortality (Logan, Nicetown–Tioga, Strawberry Mansion, Sharswood–Stanton, Mill Creek–Parkside, Cobbs Creek, Paschall–Kingsessing). Seven neighborhoods with worse crime rates than Philadelphia also have worse cancer mortality rates (Germantown, Nicetown–Tioga, Strawberry Mansion, Sharswood–Stanton, Mill Creek–Parkside, Cobbs Creek, Paschall–Kingsessing, S3 Fig). Seven neighborhoods that are worse on crime are also worse on the measure related to having access to a safe park for recreation (Nicetown–Tioga, Huntington Park–Fairhill, Upper Kensington, Juniata Park–Harrowgate, Frankford, Paschall–Kingsessing, Sharswood–Stanton). Further, there are two neighborhoods

**Table 3. Neighborhoods with overlap of worst ACS physical activity, nutrition, and smoking guidelines adherence and highest cancer mortality.**

|  | Sharswood—Stanton (Map ID: S) | Logan (Map ID: L) | Nicetown–Tioga (Map ID: N) | Strawberry Mansion (Map ID: SM) |
|---|---|---|---|---|
| #1: Body Weight | 0 | -1 | -1 | -1 |
| #2: Physical Activity | 0 | 0 | -1 | -1 |
| #3: Healthy Diet | 0 | -1 | -1 | 0 |
| #4: Alcohol Consumption | -1 | 1 | 0 | 0 |
| #5: Tobacco Use | 0 | 0 | -1 | -1 |
| #6: Supermarket Access | 0 | -1 | 0 | 0 |
| #7: Environment | -1 | 0 | -1 | -1 |

Column names indicate the neighborhoods with overlap and rows are recommendations included in guideline adherence. A value of -1 indicates the neighborhood is worse than the city overall on the recommendation, 1 indicates better, and 0 indicates no different on select recommendations where differences are observed between city and high mortality neighborhoods.

**Table 4. Neighborhoods with overlap of worst preventive services guideline adherence and highest cancer mortality.**

|  | Paschall—Kingsessing (Map ID: P) | Strawberry Mansion (Map ID: SM) | Richmond–Bridesburg (Map ID: R) |
|---|---|---|---|
| HP: PCP ratio | -1 | -1 | -1 |
| HP: Good Overall Health | -1 | -1 | 0 |
| Cancer Screening | 0 | 0 | -1 |

Column names indicate the neighborhoods with overlap and rows are recommendations included in guideline adherence. A value of -1 indicates the neighborhood is worse than the city overall on the recommendation, 1 indicates better, and 0 indicates no different on select recommendations where differences are observed between city and high mortality neighborhoods.

that are worse on crime, having a park residents feel safe visiting, physical activity, obesity, and cancer mortality (Paschall–Kingessing, Nicetown–Tioga).

The driving factors for poor adherence to Healthy People Prevention Services Guidelines in Paschall–Kingsessing and Strawberry Mansion relate to low rates of residents reporting that they are in "Good Health" and poor resident to physician ratio. In Richmond-Bridesburg, poor adherence to Healthy People Prevention Services Guidelines is driven by low access to primary care physicians and a lack of recommended screening for cancer (Table 4).

## Discussion

In this study, we generated neighborhood risk factor prevalence rates and created two composite scores to measure neighborhood-level adherence to current cancer prevention guideline recommendation set by the American Cancer Society (ACS) and Healthy People 2030 that included built environment recommendations. The first adherence score reflects ACS Physical Activity, Nutrition, and Smoking Guidelines, while the second focuses on Healthy People Prevention Services Guidelines. To our knowledge, this study is the first to measure guideline adherence at the neighborhood-level and the first to incorporate measures of the built environment in that assessment. We compared our 2 overall adherence scores to Philadelphia's geographic distribution of cancer mortality to identify neighborhoods with low guideline adherence and high cancer burden. We then determined which modifiable factors may contribute to low adherence scores in those neighborhoods in order to inform where and which type of cancer prevention interventions may be needed.

As a city, Philadelphia appears to be close to the national average or Healthy People (HP) 2030 targets for ACS recommendations related to healthy weight and healthy diet (recommendations 1 & 3). However, Philadelphia does not meet the Healthy People 2030 target on three recommendations from the overall ACS Physical Activity, Nutrition, and Smoking Guidelines adherence index. Philadelphia residents drink alcohol (recommendation #4) and smoke cigarettes (recommendation #5) at a greater rate than the target set by Healthy People 2030 (36% vs 25.4% and 20% vs 6.1%, respectively). Observing geospatial variation throughout the city, we see that the majority of the neighborhoods considered worse on alcohol binge drinking than the city overall are concentrated in Center City and South Philly, as well as two other neighborhoods in the Upper Northeast and one in Northwest Philadelphia. The neighborhoods that had worse adherence to the smoking recommendation compared to the city were in North Central, Lower Northeast, and South Philadelphia. Two neighborhoods that have high cancer mortality are worse on tobacco use (Nicetown-Tioga and Strawberry Mansion) and one is worse on alcohol consumption (Sharswood-Stanton) compared to the city overall. Philadelphia also does not meet the Healthy People 2030 target for physical activity. Neighborhoods that were worse than the city overall on this measure were in Southwest, West, North

Central, and Lower Northeast Philadelphia. Two of the high cancer mortality neighborhoods were worse on physical activity (Nicetown-Tioga and Strawberry Mansion). While alcohol and tobacco use and sedentary behavior are known risk factors for cancer mortality, many of the neighborhoods that scored worse than the city overall on these measures did not have higher cancer mortality than the city overall, likely because rates are generally unfavorable throughout Philadelphia. This suggests that in general, Philadelphia could benefit from additional efforts to reduce alcohol consumption, aid in tobacco use prevention and cessation, and increase physical activity among residents.

Regarding the Healthy People Prevention Services Guidelines, Philadelphia meets the HP targets for cancer screenings and self-report of good health. However, Philadelphia has about 220 more residents per primary care provider than the average large city in the U.S., which may influence availability and access to health care services for Philadelphia residents. This is further complicated by factors such as insurance status of residents, general health literacy, and use of Philadelphia physicians by individuals residing in the city's suburbs. Geospatially, there are two clusters of neighborhoods that are worse with regard to the population to physician measure. They are located in the Lower Northeast (Oxford Circle, Lawndale-Crescentville, Frankford, Wissinoming-Tacony, Richmond-Bridesburg), and West/Southwest (Haddington-Overbrook, Cobbs Creek, Paschall-Kingsessing) parts of the city, as well as three additional neighborhoods in the North Central (Strawberry Mansion, West Oak Lane-Cedarbrook) and far Northeastern (Torresdale North) areas, where in some neighborhoods ratios are as high as 3,611 residents per physician. This indicates that in general, access to care for Philadelphia residents should be carefully evaluated and considered as a potential risk for cancer outcomes, especially in the three neighborhoods that were worse than the city overall on primary care physician to resident ratio and cancer mortality (Richmond-Bridesburg, Strawberry Mansion, Paschall-Kingsessing).

Four of the neighborhoods with low adherence to ACS Physical Activity, Nutrition, and Smoking Guidelines also had worse cancer mortality than the city overall (Sharswood-Stanton, Logan, Nicetown-Tioga, Strawberry Mansion). Each of these neighborhoods are also worse on one of the two recommendations related to neighborhood built environment, indicating that the built environment could be an important factor to consider in cancer mortality. Three of these four (Strawberry Mansion, Sharswood-Stanton, Nicetown-Tioga) are worse on recommendation #7, which includes measures related to safety and accessibility and one neighborhood (Logan) is worse on recommendation #6, which measures supermarket access. Of the five measures related to the built environment recommendations, Philadelphia is worse than the national average on only two: car ownership (a measure of access) and violent crime (a measure of safety). However, comparing these measures with geospatial patterns of related built environment measures, such as walkability (a measure of access and exercise) and access to safe parks (a measure of safety and exercise), revealed valuable insights about how factors related to the built environment may be impacting guideline adherence in Philadelphia neighborhoods.

Philadelphia has very low vehicle ownership compared to the national average (69% vs. 92%). This could be reflective of the urban setting and high level of walkability (i.e., fewer people need a car to get around). However, as Philadelphia is one of the most impoverished big cities in the country [32], low vehicle ownership is likely also reflective of high rates of poverty within the city (fewer people can afford a car [33]). Comparing car ownership to other measures of access, we see that the neighborhoods with low car ownership and low mortality are better than the city overall on walkability, while neighborhoods with low car ownership and high mortality are not very walkable. This relationship suggests that access is a nuanced domain which may require multiple measures to capture it adequately.

Philadelphia performs worse than the nation on the built environmental measure related to violent crime. Philadelphia has a violent crime rate of approximately 2.5 times that of the U.S. overall (944 vs 379 events per 100 thousand people). Unlike car ownership, which overlaps neighborhoods that are both better and worse with regard to cancer mortality, the directionality of the overlap between crime and mortality is consistent, such that neighborhoods that are worse with regard to violent crime overlap heavily with neighborhoods that have worse mortality. Specifically, 7 of 11 neighborhoods with worse crime rates than Philadelphia also have worse cancer mortality rates, suggesting that beyond cancer prevention services, violence prevention should be an emphasis in Philadelphia and any cancer-related intervention services might also consider trauma-informed approaches when designing cancer interventions in the city, as suggested by some in oncology [34]. Further, as there is evidence that stressful life experiences are positively associated with cancer mortality [35], future researchers may explore the impact of persistent stress caused by living in neighborhoods impacted by violence on cancer health outcomes.

The spatial overlap between crime and other measures of guideline adherence included in this study warrant additional consideration. For example, while Philadelphia residents have a high access to parks within a 10-minute walk from their home (95% vs. 55% nationally [36, 37]), only 75.8% have a park they feel safe visiting within a 15-minute walk from their home. The difference between park availability and comfort visiting parks (95% vs 76%) is notable. It is possible that crime accounts for some of this difference, as our study finds that 7 of the 11 neighborhoods that are worse on access to safe parks overlap with neighborhoods with high crime. Further, as two neighborhoods that are worse on crime are also worse on having a safe park, physical activity, obesity, and cancer mortality, we assert that, in some areas of the city, crime could be impacting the availability of outdoors spaces that residents feel comfortable utilizing for exercise, thereby influencing adherence to cancer prevention guidelines, including physical activity and obesity, and possibly effecting cancer mortality.

Based on our findings we recommend that smoking, alcohol, and physical activity interventions should be implemented citywide. In the regions of North Central, West, and Southwest Philadelphia (particularly for neighborhoods listed in Table 3), crime reduction initiatives should be supported as a public health intervention, safe and affordable places to exercise should be provided, and access to healthy food and nutrition education should be increased. In North Central and Northeast Philadelphia, cancer screening and access to physicians and related measures of healthcare accessibility should be further evaluated (particularly for neighborhoods listed in Table 4).

This study is not without limitations. One limitation is that a direct comparison to the national level such as those provided in Table 1 is not always available. For example, the grocery stores variable we use looks at the ½ mile distance (which is appropriate for an urban setting), but the national statistic we use as a comparison looks at supermarkets within 1 mile (which is more appropriate for the mostly rural U.S) . While this is a limitation, we can still infer that Philadelphia is on par with the national rate, as the rate in Philadelphia is approximately half that of the national level, and the distance is also half. This same limitation applies in our measurement of recommendation #2 related to physical activity. A second limitation relates to screening measures used in the Healthy People Prevention Services Guidelines adherence score. The colorectal cancer screening measure uses only colonoscopy and does not include other applicable methods of screening, such as the fecal occult blood test, sigmoidoscopy, virtual colonoscopy, or DNA stool test. Similarly, cervical screening measure includes only PAP tests and does not include the related HPV screening. This limitation may result in an underrepresentation of colorectal or cervical cancer screening in the data used in this study. Nevertheless, this geospatial analysis at the neighborhood level can provide insights into areas

of the city that should be prioritized for secondary prevention or screening interventions. A third limitation is that we use available data and years ranges; as a result, when we compare our indices to cancer rates, the year ranges only overlap slightly at the beginning of that time frame. However, the primary purpose of the manuscript was to create an index that is reflective of modifiable risk factors over time and preliminarily test the utility of the measure by comparing poor adherence to a disease measure. Future studies should take into consideration data availability and the time frame of exposures and outcome. Lastly, this is not a causal study and the indices created herein are not intended to predict cancer outcomes. Rather, we use area-level measures and summaries to help inform cancer prevention efforts. While we find co-occurrence of high cancer mortality and low adherence in some neighborhoods and find a significant association between the two, we cannot assert that low adherence to a particular recommendation is causing a high cancer mortality in a given location. Additionally, we have demonstrated a process for generating guideline adherence indices, however, various methods could be applied that may result in somewhat different neighborhoods being identified. For example, a limitation of this analysis is that all variables are weighted similarly, when it is possible some measures at the area-level (e.g. living in an area with high smoking rates) could have more of an impact on disease outcomes than others (e.g. living >1/2 mile from a supermarket). This impact could vary depending on the outcomes studied and in the context of individual level behaviors. Therefore, findings must be mindfully interpreted, and future cancer control and prevention studies to investigate risk factor differences by neighborhood using additional sophisticated statistical testing are suggested. This could include weighting of area-level data for the generation of adherence scores, as well as multi-level analyses to better understand how the neighborhood may influence cancer prevention guideline adherence on an individual-level.

By generating summary ACS Physical Activity, Nutrition, and Smoking Guidelines and Healthy People Prevention Services Guidelines adherence scores and comparing them to cancer mortality rates, this study provides a pragmatic method for identifying neighborhoods to be prioritized for cancer interventions within a single city. This is a timely approach, as many new cancer cases are preventable through evidence-based behavioral interventions. Specifically, this study's findings suggest that the built environment (e.g., walkability, safety) may relate to guideline adherence, as supported by prior research which links neighborhood characteristics to individual risk behaviors relevant to cancer prevention. By incorporating built environment factors into a composite measure, policymakers, public health officials, and cancer centers can gain valuable insights to address the cancer burden in their respective locales. We expanded on prior county and within-city level health rankings by including measures of the built environment that have been shown to correlate with unfavorable risk behaviors associated with cancer. Tailoring interventions based on the specific recommendations driving low guideline adherence in high cancer burden areas supports a Precision Public Health approach by ensuring the efficient allocation of limited resources and has the potential to make a significant impact on cancer prevention efforts across various communities. Future studies should continue to consider the impacts of the built environment on health behaviors when prioritizing communities to target and designing interventions to address the burden of cancer.

## Supporting information

**S1 Fig. Neighborhood map of Philadelphia with each of the 46 neighborhoods included in the study outlined and labeled.**
(TIF)

**S2 Fig. Region map.** Neighborhood map (matching S1 Fig) that includes rectangles around the different regions of the city.
(TIF)

**S3 Fig. Maps of individual measures.** Series of small maps showing the neighborhood-level "better," "worse," and "no different" categories for each of the 18 measures included in the indices and for cancer mortality.
(TIF)

**S1 Table. Detailed information about included measures.** Table detailing each recommendation, where it originated, the measure(s) used to measure adherence, the data sources and years for each measure, the technique for aggregating data to the neighborhood-level, and the direction (whether being above the city overall for that measure is considered better or worse).
(DOCX)

**S2 Table. Univariate regressions predicting cancer mortality with each adherence index.** The results of two univariate regressions predicting mortality. Model 1 uses the final ACS Physical Activity, Nutrition, and Smoking Guidelines Index to predict cancer mortality rates at the neighborhood level in a linear regression. Model 2 does the same using the final Preventive Services Guideline Index. Both models are highly significant, with p-values of under 0.001 and adjusted R squares of 0.45 and 0.22, respectively.
(DOCX)

## Author Contributions

**Conceptualization:** Kari Moore, Shannon M. Lynch.

**Data curation:** Kari Moore, Heather Rollins, Kristen A. Sorice, Shannon M. Lynch.

**Formal analysis:** Tesla D. DuBois, Heather Rollins.

**Funding acquisition:** Shannon M. Lynch.

**Investigation:** Shannon M. Lynch.

**Methodology:** Tesla D. DuBois, Heather Rollins, Shannon M. Lynch.

**Project administration:** Kristen A. Sorice.

**Resources:** Shannon M. Lynch.

**Supervision:** Shannon M. Lynch.

**Validation:** John Silbaugh.

**Visualization:** Tesla D. DuBois.

**Writing – original draft:** Tesla D. DuBois.

**Writing – review & editing:** Tesla D. DuBois, Kari Moore, Heather Rollins, Shannon M. Lynch.

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
