## [Decision Letter · Decision Letter 0]

5 Apr 2024

PONE-D-23-41947Data-driven insights into neighborhood adherence to cancer prevention guidelines in PhiladelphiaPLOS ONE

Dear Dr. Lynch,

Thank you for submitting your manuscript to PLOS ONE. After careful consideration, we feel that it has merit but does not fully meet PLOS ONE’s publication criteria as it currently stands. Therefore, we invite you to submit a revised version of the manuscript that addresses the points raised during the review process.

We look forward to receiving your revised manuscript.

Kind regards,

Meghana Ray, Ph.D., MBA, B.Pharm

Academic Editor

PLOS ONE

Journal Requirements:

3. In the online submission form, you indicated that this study relies exclusively on third-party data contributed from the Philadelphia Health Management Corporation, the U.S. Census American Community Survey, the Philadelphia Police Department, the City of Philadelphia, the Trust for Public Land, the United States Department of Agriculture, and the Pennsylvania State Cancer Registry. Descriptions of variables used in this analysis, the year, and data product from which they are pulled are detailed in the supporting materials (S1 Table) of this manuscript. The Pennsylvania Cancer Registry data is available upon request directly to the Pennsylvania Department of Health. Likewise, the Southeastern Pennsylvania Household Health Survey data are available upon request to the Public Health Management Corporation. All other data sources are available to the public as part of the the Close to Home report published by the City of Philadelphia and the Cancer in Philadelphia Neighborhoods website published by Drexel University.

5. We note that Figures 2, 3, S1, S2 and S3 in your submission contain [map/satellite] images which may be copyrighted. All PLOS content is published under the Creative Commons Attribution License (CC BY 4.0), which means that the manuscript, images, and Supporting Information files will be freely available online, and any third party is permitted to access, download, copy, distribute, and use these materials in any way, even commercially, with proper attribution. For these reasons, we cannot publish previously copyrighted maps or satellite images created using proprietary data, such as Google software (Google Maps, Street View, and Earth). For more information, see our copyright guidelines: http://journals.plos.org/plosone/s/licenses-and-copyright.

a. You may seek permission from the original copyright holder of Figures 2, 3, S1, S2 and S3 to publish the content specifically under the CC BY 4.0 license.  

6. We notice that your supplementary tables are included in the manuscript file. Please remove them and upload them with the file type 'Supporting Information'. Please ensure that each Supporting Information file has a legend listed in the manuscript after the references list.

Reviewers' comments:

Reviewer's Responses to Questions

**Comments to the Author**

1. Is the manuscript technically sound, and do the data support the conclusions?

Reviewer #1: Partly

2. Has the statistical analysis been performed appropriately and rigorously? 

Reviewer #1: No

3. Have the authors made all data underlying the findings in their manuscript fully available?

Reviewer #1: Yes

4. Is the manuscript presented in an intelligible fashion and written in standard English?

Reviewer #1: Yes

5. Review Comments to the Author

Reviewer #1: Thank you for allowing me to review this interesting study. This study proposes to address the important gap of generating neighborhood-level cancer-related guideline adherence measures and indices in Philadelphia PA as an example for tracking cancer risk factors. While the question is important, the analytic choices could be improved or further justified. Most importantly, the choice of cut-off for assigning numeric value within each measure could influence final index creation substantially, and differently for each measure leading to inherent down-/up-weighting. This and more minor comments follow.

Introduction

1. Lines 60-62. ‘nearly 20% of all new cancers diagnosed are attributable to modifiable risks’, the reference supporting this cannot be found and 20% is lower than many other published estimates (Islami et al CA a cancer journal for clinicians 2018, GBD 2019 Cancer Risk Factors Team Lancet 2022, Collatuzzo and Boffetta Ann Rev Pub Health 2023). Consider revising.

2.

Methods

3. Cancer mortality rates from 2012-2016. It is unclear why the ‘outcome’ cancer data is dated prior to the ‘exposures’ when cancer data presumably exists post-2018 or most exposure data pre-2012.

4. Creation of indices. Given availability of cancer burden attributable risk estimates for many of these measures, it is unclear why each is given equal weight in the index creation; e.g., tobacco is well accepted to contribute substantially to cancer mortality, while >½ mile from a supermarket likely contributes far less to cancer mortality.

5. Creation of indices. Please provide justification for why sugar sweetened beverage consumption is part of body weight recommendation when it is not part of body weight in the original ACS recommendation (doi.org/10.3322/caac.21719), rather it’s part of diet.

6. Creation of indices. I having a hard time understanding calculation of the vehicle access measure involving MOE. Are MOE’s of Philly and Philly neighborhoods being compared to one another? Or the ACS mean estimates Philly and Philly neighborhoods of vehicle access being compared to one another while accounting for MOEs to generate 95% CIs? If the latter, I’m still confused where z-scores come into the calculation and how the vast majority of neighborhoods appear to be outside the |1.96| range of the z-score. Can you please clarify?

7. Creation of indices. I am concerned that Step 2 recommendation z-score, and therefore, step 3 overall adherence scores based on sums of step 2 z-scores is highly sensitive to step 1 choice of cut-off for better/worse/no different of each measure. For example some measures will generate a 1/better or -1/worse based on quartiles of the 46 neighborhoods; guaranteeing that the measure will be 1/better for ~12 neighborhoods and -1/worse for another ~12 neighborhoods. However, other measure’s assignment of 1/better or -1/worse is based on exceedance of 95% credible intervals (>97.5% or < 2.5%). Predictably, those credible interval-based measures yield far fewer 1s and -1s, and therefore, are essentially down-weighted in creation of the overall adherence scores. This should be addressed ideally through creation of measures that are not inherently weighted differently by assignment of values (1,0,-1), and/or sensitivity analyses with various cutoff values assigned to each measure and results reported in the supplement. Noting as a limitation in the Discussion is correct, but this limitation is addressable within the scope of this study’s purpose and created by the study design itself.

8. Figure 3. Why were maps limited to neighborhoods identified as high mortality and low/worse adherence, as opposed to calculating and/or displaying a correlation – spatial/LISA or aspatial/spearman - between the measures? If the goal is to use these indices for neighborhood-level tracking of cancer related risk factors, then it would be important to know if they correlate with neighborhood level cancer burden. If they do not, then their utility seems diminished.

Results

9. Temporal mismatch and causal language. Related to comment 3 above, relationships between the temporally mismatched variables are described using ‘coincided’, ‘driver’, ‘influencing’. Consider strengthening methods – temporally aligned data, causal inference, etc. - to justify causal language or describe relationships more appropriately.

10. S3 10 min walk to park. As a quartile measure, how are there no better/quartile 4 neighborhoods?

6. PLOS authors have the option to publish the peer review history of their article (what does this mean?). If published, this will include your full peer review and any attached files.

Reviewer #1: **Yes: **Jesse Plascak

---

## [Author Response · Author response to Decision Letter 0]

27 Jun 2024

Thank you to the reviewers. Please see response to reviewer attachment.

---

## [Decision Letter · Decision Letter 1]

23 Oct 2024

Data-driven insights into neighborhood adherence to cancer prevention guidelines in Philadelphia

PONE-D-23-41947R1

Dear Dr. Lynch,

We’re pleased to inform you that your manuscript has been judged scientifically suitable for publication and will be formally accepted for publication once it meets all outstanding technical requirements.

Kind regards,

Weijun Yu, Ph.D., M.D., M.S.

Academic Editor

PLOS ONE

Additional Editor Comments (optional):

Reviewers' comments:

Reviewer's Responses to Questions

**Comments to the Author**

1. If the authors have adequately addressed your comments raised in a previous round of review and you feel that this manuscript is now acceptable for publication, you may indicate that here to bypass the “Comments to the Author” section, enter your conflict of interest statement in the “Confidential to Editor” section, and submit your "Accept" recommendation.

Reviewer #1: All comments have been addressed

Reviewer #2: All comments have been addressed

2. Is the manuscript technically sound, and do the data support the conclusions?

Reviewer #1: Yes

Reviewer #2: Yes

3. Has the statistical analysis been performed appropriately and rigorously? 

Reviewer #1: Yes

Reviewer #2: Yes

4. Have the authors made all data underlying the findings in their manuscript fully available?

Reviewer #1: Yes

Reviewer #2: Yes

5. Is the manuscript presented in an intelligible fashion and written in standard English?

Reviewer #1: Yes

Reviewer #2: Yes

6. Review Comments to the Author

Reviewer #1: Thank you for being so open to my suggestions and congratulations on a study that will be helpful to the cancer prevention and control community.

Only one minor wording suggestion: “Recommendation #7 includes four measures. Philadelphia has a high walkability score compared to other cities with at least 200 thousand …” Consider revising 200 thousand to 200,000.

Reviewer #2: The authors have fully addressed and resolved all the concerns raised during the initial review. The revisions have substantially improved the manuscript, and I am satisfied that the authors have made the necessary changes to meet the publishing standards of PLOS ONE. The study presents valuable insights that will benefit PLOS ONE's readership.

Given the thoroughness of the revisions and the overall quality of the manuscript, I recommend that this study be considered for publication.

7. PLOS authors have the option to publish the peer review history of their article (what does this mean?). If published, this will include your full peer review and any attached files.

Reviewer #1: **Yes: **Jesse Plascak

Reviewer #2: No

---

## [Editor Report · Acceptance letter]

8 Nov 2024

PONE-D-23-41947R1 

PLOS ONE

Dear Dr. Lynch, 

I'm pleased to inform you that your manuscript has been deemed suitable for publication in PLOS ONE. Congratulations! Your manuscript is now being handed over to our production team.

Kind regards, 

on behalf of

Dr. Weijun Yu 

Academic Editor

PLOS ONE